# Suitable Disinfectants with Proven Efficacy for Genetically Modified Viruses and Viral Vectors

**DOI:** 10.3390/v15112179

**Published:** 2023-10-30

**Authors:** Maren Eggers, Ingeborg Schwebke, Johannes Blümel, Franziska Brandt, Helmut Fickenscher, Jürgen Gebel, Nils Hübner, Janis A. Müller, Holger F. Rabenau, Ingrid Rapp, Sven Reiche, Eike Steinmann, Jochen Steinmann, Paula Zwicker, Miranda Suchomel

**Affiliations:** 1Laboratory Prof. Dr. G. Enders MVZ GbR, Rosenbergstr. 85, 70193 Stuttgart, Germany; 2Expert Committee on Virus Disinfection of the German Association for the Control of Viral Diseases (DVV) e.V. and the Society for Virology (GfV) e.V., 69126 Heidelberg, Germany; inge.schwebke@gmx.de; 3Paul-Ehrlich-Institute, Department of Virology, Paul-Ehrlich-Straße 51-56, 63225 Langen, Germany; johannes.bluemel@pei.de; 4Federal Institute for Drugs and Medical Devices, Kurt-Georg-Kiesinger-Allee 3, 53175 Bonn, Germany; franziska.brandt@bfarm.de; 5Institute for Infection Medicine, Christian-Albrechts-University Kiel, University Clinic Schleswig-Holstein, Bruinswiker Straße 4, 24105 Kiel, Germany; fickenscher@infmed.uni-kiel.de; 6VAH c/o Institute for Hygiene and Public Health, Venusberg-Campus 1, 53127 Bonn, Germany; juergen.gebel@ukbonn.de; 7Institute of Hygiene and Environmental Medicine, University Medicine Greifswald, W. Rathenaustr. 49, 17475 Greifswald, Germany; nils.huebner@med.uni-greifswald.de (N.H.); paula.zwicker@med.uni-greifswald.de (P.Z.); 8Institute of Virology, Hans-Meerwein Straße 2, 35043 Marburg, Germany; janismueller@uni-marburg.de; 9Institute for Medical Virology, University Hospital, Goethe University Frankfurt am Main, 60596 Frankfurt, Germany; rabenau@em.uni-frankfurt.de; 10Boehringer Ingelheim Therapeutics GmbH, Beim Braunland 1, 88416 Ochsenhausen, Germany; ingrid.rapp@boehringer-ingelheim.com; 11Friedrich-Loeffler-Institute, Federal Research Institute for Animal Health, Department of Experimental Animal Facilities and Biorisk Management, Suedufer 10, 17493 Greifswald-Insel Riems, Germany; sven.reiche@fli.de; 12Department for Molecular & Medical Virology, Ruhr University Bochum, 44801 Bochum, Germany; eike.steinmann@ruhr-uni-bochum.de; 13Dr. Bill + Partner GmbH Institute for Hygiene and Microbiology, Norderoog 2, 28259 Bremen, Germany; jochen-steinmann@web.de; 14Institute of Hygiene and Applied Immunology, Medical University of Vienna, Kinderspitalgasse 15, 1090 Vienna, Austria; miranda.suchomel@meduniwien.ac.at; 15Austrian Society for Hygiene, Microbiology and Preventive Medicine (ÖGHMP) c/o MAW, Freyung 6/3, 1010 Vienna, Austria

**Keywords:** genetically modified organisms (GMOs), viral vectors, virucidal activity, suspension tests, virus inactivation, disinfection, AAV, vector

## Abstract

Viral disinfection is important for medical facilities, the food industry, and the veterinary field, especially in terms of controlling virus outbreaks. Therefore, standardized methods and activity levels are available for these areas. Usually, disinfectants used in these areas are characterized by their activity against test organisms (i.e., viruses, bacteria, and/or yeasts). This activity is usually determined using a suspension test in which the test organism is incubated with the respective disinfectant in solution to assess its bactericidal, yeasticidal, or virucidal activity. In addition, carrier methods that more closely reflect real-world applications have been developed, in which microorganisms are applied to the surface of a carrier (e.g., stainless steel frosted glass, or polyvinyl chloride (PVC)) and then dried. However, to date, no standardized methods have become available for addressing genetically modified vectors or disinfection-resistant oncolytic viruses such as the H1-parvovirus. Particularly, such non-enveloped viruses, which are highly resistant to disinfectants, are not taken into account in European standards. This article proposes a new activity claim known as “virucidal activity PLUS”, summarizes the available methods for evaluating the virucidal activity of chemical disinfectants against genetically modified organisms (GMOs) using current European standards, including the activity against highly resistant parvoviridae such as the adeno-associated virus (AAV), and provides guidance on the selection of disinfectants for pharmaceutical manufacturers, laboratories, and clinical users.

## 1. Introduction

In Europe, the spectrum of the efficacy of disinfection processes against viruses currently includes the following three activity levels [1,2], which are “active against enveloped viruses”, “limited spectrum of virucidal activity”, and “virucidal activity” (which includes both activity against enveloped viruses and a limited spectrum of virucidal activity). Depending on the risk assessment and area of application of the disinfectant (hands, surfaces, instruments, and laundry), different activity levels are required and can be declared on the basis of existing standardized test methods. For hygienic hand disinfection and surface disinfection, the required activity levels are “active against enveloped viruses”, “limited spectrum of virucidal activity”, and “virucidal activity”. For instrument disinfection they are “active against enveloped viruses” and “virucidal activity”, and for laundry disinfection it is “virucidal activity”.

The above-mentioned activity levels were originally developed for the medical area. For other areas, additional terms with different content are under discussion in the technical committee (TC) 216 of the European Committee for Standardisation (CEN, Comité Européen de Normalisation in French).

In addition to the efficacy of disinfection processes against viruses, this overview draws attention to genetically modified organisms (GMOs) of viral origin. They are used in numerous laboratories for very different purposes, e.g., for basic research (viral transduction), for research on and the production of vaccines (e.g., recombinant vaccine viruses), for gene transfer (viral vectors), or for direct tumour therapy (e.g., native oncolytic viruses (OV) such as H1-Parvovirus or recombinant oncolytic viruses such as the approved 2015 Talimogene Laherparepvec based on HSV-1) [3,4]. OVs include numerous virus families that exhibit natural tropisms for specific extracellular oncogenic receptors or have been genetically engineered to display targeted tropisms for tumour cells while being restricted in their effects on normal cells [5]. Furthermore, engineered OVs often additionally express immunotherapeutic factors. In contrast to gene therapy vectors, which are used to transfer therapeutic genetic material to target tissues and cannot actively infect the host cells, OVs are less attenuated and can replicate in infected tissues [6].

The continuous progress of research in this field and the constantly evolving technical possibilities have also led to the development of new perspectives and applications for cell- and gene-based medicines. For example, viral vectors are successfully used in the production of functionally modified T cells with chimeric antigen receptors (CAR-T cells) [7] or offer promising tools for the treatment of genetic diseases. Other gene therapy treatments have been designed to treat haemophilia A, haemophilia B, or patients with inherited rare diseases [8,9]. However, in the wake of the COVID-19 pandemic, there is a public perception and a growing trend of scepticism towards research institutions regarding poor lab safety standards or even lab leaks due to inadequate biosafety levels. Viral vectors or genetically modified viruses are assigned to different genetic safety levels. Initially, this occurs independently of the type and function of the nucleic acid insertions. In Germany, the classification of viral vectors or GMOs is carried out by the Central Commission for Biological Safety (ZKBS) on the basis of a risk assessment of the GMOs or viral vectors to determine their safety for humans, animals and the environment. Viral vectors are classified as GMOs. As [10] the ZKBS points out, organisms are not only biological entities that are capable of reproducing but also transmitting genetic material. In addition to the typical GMOs of retroviral, adenoviral, or parvoviral origin, oncolytic viruses that are not genetically modified, such as H-1 parvovirus, are also the subject of this overview.

In the production and use of GMOs or native oncolytic viruses, the question arises as to which disinfectants should be used when handling them (e.g., in research, development, production, or clinical application). The selection of a suitable disinfectant cannot be derived from genetic safety classification, as this indicates potential risks to other organisms and the environment but does not consider the tolerance or sensitivity of the GMOs or native oncolytic viruses to disinfectants.

The aim of this overview is to make the various aspects of efficacy testing and the spectrum of antiviral activity in the selection of disinfectants transparent for supervisory and monitoring authorities, biosafety officers, and operators of genetic engineering facilities. In this context, we primarily consider activity against viruses and address the anticipated changes caused by the European biocide legislation.

## 2. Efficacy of Disinfectants Depends on Viral Structure

The infectivity of virus particles is essentially determined by the nature of the viral envelope or the viral capsid (in the case of non-enveloped viruses). Their basic biological structure can be derived from their viral families or from the respective “donor and recipient organisms” that determine their biochemical and physical properties. Since disinfectants initially attack the viral envelope and/or the capsid of virus particles, the selection of the disinfectant for recombinant viruses or viral vectors (such as chimeric viruses) depends on the origin of the viral envelope or the viral capsid. Enveloped recombinant viruses such as viral vectors (GMOs) or “infectious” nucleic acids are unique cases since GMOs often contain modified viral proteins or other additionally expressed proteins in their viral membrane. Nevertheless, these genetically modified proteins are not expected to significantly alter the stability of the lipid envelope when exposed to disinfectants.

Comparative studies with a large number of enveloped viruses have shown that vaccinia virus—the model virus for enveloped viruses—still has the highest stability and is thus a good benchmark [11,12,13,14] and a well-suited model virus for enveloped viruses. Its stability might be higher than that of others, as the cell-associated enveloped virus (CEV) and the extracellular enveloped virus (EEV) are both surrounded by two membrane layers, while the intracellular enveloped virus (IEV) has three envelopes [15]. Aside from the structural proteins and lipids, viruses contain nucleic acid that can still be considered as infectious. However, not all disinfectants destroy or inactivate the nucleic acids (DNA or RNA) contained in the virus particles. Certain virus groups (e.g., vaccinia viruses, herpes viruses, adenoviruses, simian virus 40 (SV40), picornaviruses, or parvoviruses) possess so-called “infectious” virus genomes. In principle, progeny viruses can develop from these genomes if they enter or are introduced into suitable host cells. However, eukaryotic cells do not have natural mechanisms for the uptake of free (non-encapsulated) viral genomes. Therefore, extremely high concentrations and special technical procedures are required to introduce free nucleic acids into cells in sufficient quantities for productive replication. Therefore, the infectivity of isolated viral genomes plays only a minor role compared to the infectivity of complete viral particles in the context of the potential risk of infection they pose.

According to Klein and DeForest [16], the activity level for the declaration of disinfectants is derived from the virus structure [1,17]. Disinfectants that are “active against enveloped viruses” are sufficient for use against all enveloped viruses. Within the group of non-enveloped viruses, adenoviruses, noroviruses, and rotaviruses form the group of lipophilic non-enveloped viruses, for which products with the activity level “*limited spectrum of virucidal activity*” are used. For the more hydrophilic non-enveloped virus agents, the activity level “*virucidal activity*” is required. Correspondingly, the same applies mutatis mutandis to GMOs: GMOs or native oncolytic viruses based on enveloped viruses can be inactivated using disinfectants that are “*active against enveloped viruses*”. In the case of more lipophilic non-enveloped viruses—rotaviruses, noroviruses, adenoviruses or GMOs derived from them—disinfectants are required with at least the “limited spectrum of virucidal activity” activity level [2]. However, some GMOs or native oncolytic viruses based on hydrophilic non-enveloped viruses such as parvoviruses are not inactivated by products with a “virucidal activity” level. Thus, we require a disinfectant with activity against parvoviruses [18,19]. Therefore, the German Committee of Virus Disinfection (DVV/GfK) and the Disinfectant Commission of the VAH (Association of Applied Hygiene) [20] have implemented a new virucidal activity level “*virucidal activity PLUS*”, requiring a practical test against parvoviridae such as those described in the German guideline of 2012 for the quantitative evaluation of the virucidal activity of chemical disinfectants on non-porous surfaces [21].

## 3. Test Methods for the Declaration of Activity against Viruses

For the evaluation of the virucidal efficacy of disinfectants, a two-step procedure is provided at a national [22,23,24,25] and a European level [26,27,28,29,30,31]. In this model, the efficiency must be confirmed in quantitative suspension tests (phase 2, step 1) followed by tests under practical conditions (phase 2, step 2). In suspension tests, virus particles in solution are mixed with disinfectant plus interfering substances to simulate organic solutions and the remaining titres determined via endpoint titration. This generally determines whether a substance reduces viral infectivity and is thus active against viruses. Suspension tests are well suited for use in preliminary investigations but are carried out with a considerable excess of disinfectants, thus simulating conditions that are usually not present in practice. Therefore, the use of suspension tests often results in activities at lower concentrations and/or shorter exposure times compared to the tests performed under practical conditions (phase 2, step 2) and thus might overestimate efficacy.

Therefore, the tests performed under practical conditions must correspond to the situation in which the disinfectants are to be used, e.g., for hand hygiene, spraying surfaces, and immersing instruments. However, to ensure the efficacy of disinfectants, the test methods must not only simulate the type of application but should also consider possible organic contamination. At present, coordinated phase 2, step 2 tests are only used at the European level to assess antiviral efficacy for surface disinfection via the spray method (application without mechanical action, EN 16777) [28] and for chemical and chemo-thermal instrument disinfection via the immersion method (EN 17111) [29]. However, this is not the case for surface disinfection via mechanical action or hand disinfection. For this reason, viral test methods are currently being developed for application to the various areas of application, which also include the different areas of effect, by specifying the respective test viruses.

Specifically, hand disinfectants must be tested on the hands of volunteers and the achieved reduction must be compared to that achieved using a reference product tested in parallel on the same study subjects [32]. For surface disinfectants, the tests must consider whether the surfaces are wiped (i.e., disinfection is carried out with a mechanical component) or whether the product is sprayed on (without mechanical action). Instrument disinfectants are normally analyzed for their efficacy with immersion procedures.

The desired efficacy of products can only be guaranteed if they are also used as specified in the respective application-specific test. To date, test methods have been developed for the medical and veterinary fields and for the food sector. Special test methods have not yet been developed for the laboratory sector. If surfaces or the hands of employees are to be disinfected in laboratories, however, products with proven efficacy can be used in accordance with the test methods appropriate to the medical or veterinary field. But if the products are to be used in a way that deviates from the established applications, additional tests should be performed that largely take into account the desired use.

If there are no official national lists for disinfectants with proven virucidal activity, or if the existing lists do not include virus-active products (i.e., [33,34,35]), the user must rely on the label from the manufacturer.

In conclusion, the claim or labelling of products as being “*active against enveloped viruses*”, or as displaying “*limited spectrum of virucidal activity*” and “*virucidal activity*”, does not mean that products have undergone uniform tests and have actually met the requirements for the confirmation of efficacy, especially in the application setting.

## 4. Efficacy Level of Disinfectants and Surrogate Viruses for Disinfectant Testing in Germany and Europe

In Germany, virucidal claims such as “*active against enveloped viruses*”, “*limited spectrum of virucidal activity*”, or “*virucidal activity*” are in principle harmonized with those in the rest of Europe. However, the proof of disinfectant efficacy in Germany is based on more extensive tests than those used in Europe as additional surrogate viruses (see tables) and reproducibility (i.e., more test runs) are required. Table 1a–c list the requirements for virus-active products in the medical field.

As shown in Table 1a (*active against enveloped viruses*) and Table 1b (*limited spectrum of virucidal activity*), there is little to no difference in the test viruses. While the European method [27] uses vaccinia virus as the test virus in the quantitative suspension test (phase 2/step 1), the German methods [22,23] additionally use bovine viral diarrhoea virus (BVDV) when testing oxidative products with the activity level “*active against enveloped viruses*”.

In terms of “*virucidal activity*” (Table 1c), enteroviruses (EV) such as EV71 and EVD68 and poliovirus are covered by both the German and the European methods. However, this claim does not cover the highly resistant outbreak viruses hepatitis A (HAV) and hepatitis E (HEV). Also not included are parvoviridae, including adeno-associated viruses (AAV) used in gene therapy. Recently, clusters of AAV type 2 were globally reported to occur in children with acute severe hepatitis in 2022 [38,39]. An additional test virus simian virus (SV) 40 (as surrogate for human papillomavirus (HPV)) in the DVV/RKI test method makes another relevant clinical difference as this oncogenic virus is not included in the European standards.

Products with activity against enveloped viruses remain the correct choice for all indications where viruses are enveloped. Additionally, the most common non-enveloped viruses need to be adequately targeted in the area of the application of surface disinfection, including the disinfection of medical devices.

Since parvoviruses are particularly resistant to disinfectants, products that are to be used against parvoviridae, including AAV, native oncolytic parvoviruses, HAV and HEV, must also have proof of activity against parvoviruses. This requires that additional tests be performed with parvoviruses. However, the current European standards for testing under practical conditions [28,29] and drafts of new standards published in recent years do not include murine parvovirus (MVM) as a test organism mandated for testing under the practical conditions of surface and instrument disinfectants (with the exception of chemo-thermal instrument disinfection).

The routine use of murine parvovirus as a test organism is problematic because the active ingredient concentration or exposure time must be significantly increased to demonstrate sufficient efficacy against parvoviruses and those active ingredients are limited.

Therefore, for routine disinfection, the previous virucidal claim remains valid, but a new claim needs to be implemented in the EU to define virucidal activity against highly resistant viruses. As in the case of Germany [17], this could be the implementation of either “*virucidal activity PLUS*” or “*high level virucidal active*” for highly resistant viruses.

Table 2 presents a proposal for a new virucidal activity claim (“*virucidal activity PLUS*”) as already introduced in Germany.

## 5. Disinfectant for Oncolytic Viruses or Viral Vectors in Therapy

For users of disinfectants, it is not easy to select suitable conditions or products based on the information available regarding the various test methods. For example, parvoviruses are intrinsically resistant to alcoholic formulations [18,37,40]. This high tolerance to alcohols was also found for HEV [37] and HAV [41,42]. Therefore, hand disinfectants are not available against parvovirus-based GMOs or native oncolytic parvoviruses, nor are they produced for HAV. For hand hygiene, wearing protective gloves and hand washing is recommended such as in the case of bacterial spores [43,44]. Note that hand washing can only mechanically reduce the viral load from the hands as soaps are not active against these resilient viruses. In contrast, for products based on peracetic acid, aldehydes, oxygen scavengers, and chloramine-T, activity against murine parvovirus was confirmed in practical tests in various testing laboratories, leading some to claim “*virucidal activity PLUS*” for surface disinfection without mechanical action [19,45].

For users of disinfectants, it is not easy to select suitable conditions or products based on the information available regarding the various test methods. For example, parvoviruses are intrinsically resistant to alcoholic formulations [18,37,40]. This high tolerance to alcohols was also found for HEV [37] and HAV [41,42]. Therefore, hand disinfectants are not available against parvovirus-based GMOs or native oncolytic parvoviruses, nor are they produced for HAV. For hand hygiene, wearing protective gloves and hand washing is recommended such as in the case of bacterial spores [43,44]. Note that hand washing can only mechanically reduce the viral load from the hands as soaps are not active against these resilient viruses. In contrast, for products based on peracetic acid, aldehydes, oxygen scavengers, and chloramine-T, activity against murine parvovirus was confirmed in practical tests in various testing laboratories, leading some to claim “*virucidal activity PLUS*” for surface disinfection without mechanical action [19,45].

Based on the different virucidal activity levels as described in Table 1a–c and Table 2, Table 3 and Table 4 assign the corresponding disinfectant levels of activity to the most commonly used viral GMOs.

It is evident that an oncolytic herpesvirus should not differ from a field herpesvirus regarding its sensitivity to disinfectants as the most important point in this is the envelope, i.e., a lipid bilayer. Just as with pseudo- or hybrid viruses, where, e.g., VSV-G protein is incorporated into a lentivirus, characteristics such as tropism might be changed, but the new virus retains a lipid envelope that can easily be inactivated by disinfectants. Therefore, a lentiviral vector in gene therapy should be as susceptible to disinfectants as HIV. The use of nonenveloped viruses with a completely new capsid structure due to modifications of the capsid proteins can eventually lead to the production of viral-based vectors, a variety of viral cage with a completely different symmetry and, therefore, with different gaps generated between capsomers. These differences may increment or reduce the possibility of disinfectants acting and always must be characterized virologically first (tropism, etc.) and with regard to their susceptibility to disinfectants. However, so far, the authors are not aware of any virus in which the biophysical properties have been altered to the point that disinfectants are no longer active. Note, the possible assignment of reoviruses is still under discussion. As long as no extensive data on the tenacity of reoviruses are available, disinfectants should be used with the virucidal activity claim of action according to European standards. Despite this limitation, the information provided in this review is relevant for users and authorities as genetically modified viruses are used in research and medicine. Additionally, the knowledge of how to inactivate them must be made available.

Table 4 also takes into account helper viruses. These are used in the production of some viral vectors, and some of their structural components are required for the replication of the viral vector.

## 6. Future Aspects Due to European Biocide Regulation

The implementation of the European Biocide Regulation No. 528/2012 [46] is unsettling the monitoring authorities and genetic engineering facilities in Europe, as many well-established products may or will no longer be available on the market. Nowadays, all biocidal products require authorization before they can be placed on the market, and the active substances contained in that biocidal product must be previously approved. The Biocidal Products Regulation (BPR) classifies the biocidal products into 22 biocidal product-types, with surface disinfectants belonging to product type (PT) 2 “Disinfectants and algaecides not intended for direct use on humans or animals”. This product type contains, in addition to disinfectants used in the medical field or in laboratories, products for completely different applications such as the treatment of bath water, air conditioning systems, or soils (i.e., sand, soil). However, such a collective group makes it difficult to clearly delineate the areas of application and thus to select suitable products, e.g., for laboratories.

At present, the majority of disinfectants are marketable on the basis of transitional regulations, but they do not yet have authorization as biocidal products. The authorization of biocidal products includes not only efficacy testing but also the assessment of risks to humans and the environment and is carried out in a two-step process. First, the active substances contained in the biocidal products are evaluated using a European procedure. Active substances that have been positively evaluated (“approved”) are included in the so-called Union list. This list results in the deadlines during which the authorization of the product must be applied for. If no application is submitted, product marketability ends after a fixed period. Against this background, impacts on the availability of disinfectants and potentially on their operational conditions are already present or expected as the majority of disinfectants are subject to this regulation. As an example, Greek authorities have proposed to classify ethanol, which has been reliable as a disinfectant in everyday use, as a “CMR” (carcinogenic, mutagenic, and toxic for reproduction) [47], which would severely affect its use and thus eliminate a safe, available, and inexpensive ingredient.

## 7. Conclusions

As has been shown, a variety of test methods exist at both the national and European level that can be used for the evaluation of disinfectants with declared virucidal activity. In addition to the methods and claims described here for use in the human medical area, there are also existing national and European methods that can be applied to the veterinary field or food, industrial, domestic, and institutional issues. However, there are still no methods or specifications for laboratories working with viral GMOs or oncolytic viruses available. Additionally, highly resistant parvoviridae, HAV and HEV, are not yet considered for surface disinfection in European standards. Also, practical test methods for virucidal efficacy are currently not available for all applications such as surface, instrument, textile, or hand disinfection, although much has been implemented in recent years [48]. However, as specified in EN 14885 [26], further tests under additional conditions such as test organisms, temperature, contact time, and interfering substances should be carried out according to the claimed use of a disinfectant. Therefore, EN 14885 enables tests with a highly resistant test virus such as parvovirus, using a modified standard to provide an opportunity to test disinfectants against resistant GMO or oncolytic parvoviruses. This could be the basis for the new claim “*virucidal PLUS*”.

In Europe, the labelling of the virucidal efficacy of products by the manufacturer is not uniformly regulated for medicinal products and biocides [49]. In both cases, however, only the area of application (e.g., food, medical, institutional) and the instructions for use must be indicated on the product label, but not the test methods (if available) on which the virucidal activity level is based. Unfortunately, as long as no practical tests for surface disinfection with mechanical action (four field test) or hand disinfection are approved at the European level, a manufacturer can state and advertise a “*virucidal activity*” of such products even though it has only undergone a quantitative suspension test.

Taken together, the recommendation for and selection of suitable disinfectants with proven efficacy for GMO and viral vectors is complex. Since the evaluation of laboratory test reports for the selection or comparison of different possible disinfectants with proven virucidal efficacy requires special knowledge and extensive experience, access to accordingly reviewed lists of disinfectants with proven efficacy for the different levels may be helpful for both the user and for supervisory or monitoring authorities.

## Figures and Tables

**Table 1 viruses-15-02179-t001:** (**a**) Overview of disinfectant activity level “*active against enveloped viruses*” and associated test methods for the human medical area. (**b**) Overview of disinfectant activity level “*limited spectrum of virucidal activity*” and associated test methods for the human medical area. (**c**) Overview of disinfectant activity level “*virucidal activity*” and associated test methods for the human medical area.

(**a**)		
**Active against**	Enveloped viruses (e.g., HBV, HCV, HIV, influenza viruses, herpes viruses)	
**Basis of the declaration**	**Test methods and associated test viruses**	
	DVV/VAH	European standards
**Quantitative suspension test** **(Phase 2/Step 1)**	DVV/RKI Guideline [22,23] EN 14476 [27]	EN 14476 [27]
	Vaccinia virus * BVDV (oxidative products)	Vaccinia virus *
**Practical test:** **Surface disinfection** **(Phase 2/Step 2)**	DVV Guideline [21] EN 16777 [28]	EN 16777 [28]
	Vaccinia virus *	Vaccinia virus *
**Practical test: Instrument disinfectio** **(Phase 2/Step 2)**	EN 17111 [29] (products for pre-cleaning with a combined cleaner/disinfectant)	
	Vaccinia virus *	Vaccinia virus *
(**b**)		
**Active against**	Enveloped viruses and lipophilic non-enveloped viruses (adenoviruses, noroviruses and rotaviruses)	
**Basis of the declaration**	**Test methods and associated test viruses**	
	DVV/VAH	European standards
**Quantitative suspension test** **(Phase 2/Step 1)**	DVV/RKI Guideline [22,23] EN 14476 [27]	EN 14476 [27]
	Adenovirus MNV	Adenovirus MNV
**Practical test:** **Surface disinfection** **(Phase 2/Step 2)**	DVV Guideline [21] EN 16777 [28]	EN 16777 [28]
	Adenovirus MNV	Adenovirus MNV
**Practical test: Instrument disinfection** **(Phase 2/Step 2)**	Claim cannot be issued	

(**c**)		
**Active against**	Enveloped and non-enveloped viruses (except parvoviridae such as AAV, HAV, HEV)	Enveloped and non-enveloped viruses (except parvoviridae such as AAV, HAV, HEV, papillomaviruses)
**Basis of the declaration**	**Test methods and associated test viruses**	
	DVV/VAH	European standards
**Quantitative suspension test** **(Phase 2/Step 1)**	DVV/RKI Guideline [22,23] EN 14476 [27]	EN 14476 [27]
	Adenovirus MNV Poliovirus ^†^ SV40 [36] MVM (instrument disinfection ≥ 40 °C)	Adenovirus MNV Poliovirus MVM (instrument disinfection ≥ 40 °C)
**Practical test:** **Surface disinfection** **(Phase 2/Step 2)**	DVV Guideline [21] EN 16777 [28]	EN 16777 [28]
	Adenovirus MNV	Adenovirus MNV
**Practical test: ** **Instrument disinfection** **(Phase 2/Step 2)**	EN 17111 [29]	
	Adenovirus (<40 °C) MNV (<40 °C) SV40 (<40 °C) MVM (≥40 °C)	Adenovirus (<40 °C) MNV (<40 °C) MVM (≥40 °C)

* Modified vaccinia virus Ankara (MVA) or vaccinia virus strain Elstree. HBV, hepatitis B virus; HCV, hepatitis C virus; HIV, human immunodeficiency virus; BVDV, bovine viral diarrhoea virus. MNV, murine norovirus. AAV, adeno-associated virus; HAV, hepatitis A virus; HEV, hepatitis E virus; SV40, simian virus 40; MVM, murine parvovirus (minute virus of mice; rodent protoparvovirus 1). ^†^ Although poliovirus is usually the most resistant test virus in the suspension test, it cannot be used for practical tests due to its loss in titre after drying [19,37].

**Table 2 viruses-15-02179-t002:** Proposal for a disinfectant activity level “*virucidal activity PLUS*” and associated test methods for the human medical area.

**Active against**	enveloped and non-enveloped viruses (incl. parvoviridae such as AAV, HAV, HEV, papillomaviruses)	Claim cannot be issued yet
**Basis of the declaration**	**Test methods and associated test viruses**	
	DVV/VAH	European standards
**Quantitative suspension test** **(phase 2/step 1)**	DVV/RKI Guideline [22,23] EN 14476 [27]	n.a.
	Adenovirus MNV Poliovirus SV40	n.a.
**Practical test: ** **Surface disinfection** **(phase 2/step 2)**	DVV Guideline [21]	n.a.
	Adenovirus MNV MVM	n.a.

AAV, adeno-associated virus; HAV, hepatitis A virus; HEV, hepatitis E virus; MNV, murine norovirus; SV40, simian virus 40; MVM, murine parvovirus (minute virus of mice; rodent protoparvovirus 1). n.a. not applicable.

**Table 3 viruses-15-02179-t003:** Selected oncolytic viruses and their required disinfectant activity levels.

Oncolytic Virus	Virus Family/ Group	Replication-Competent	Disinfectant Activity Level to be Used
Recombinant adenovirus (GMO)	*Adenoviridae*	yes	*limited spectrum of virucidal activity*
Herpes simplex virus type 1 (GMO or no GMO)	*Herpesviridae*	yes	*active against enveloped viruses*
Newcastle disease virus	*Paramyxoviridae*	yes	*active against enveloped viruses*
Recombinant measles virus (GMO)	*Paramyxoviridae*	yes	*active against enveloped viruses*
VSV	*Rhabdoviridae*	yes	*active against enveloped viruses*
Zikavirus	*Flaviviridae*	yes	*active against enveloped viruses*
Poliovirus	*Picornaviridae*	yes	*virucidal activity*
Parvovirus H-1	*Parvoviridae*	yes	*virucidal activity PLUS*
recombinant vaccinia virus (GMO)	*Poxviridae*	yes	*active against enveloped viruses*
recombinant fowlpox virus (GMO)	*Poxviridae*	yes	*active against enveloped viruses*
Reovirus 3	*Reoviridae*	yes	*virucidal activity* *
Senecavirus A	*Picornaviridae*	yes	*virucidal activity* *
Coxsackievirus A21	*Picornaviridae*	yes	*virucidal activity* *
Poliovirus type 1 Sabin (GMO)	*Picornaviridae*	yes	*virucidal activity* *

* virucidal activity according to European standards is sufficient.

**Table 4 viruses-15-02179-t004:** Recombinant viral vectors for gene therapy and vaccines as GMOs and the activity level of disinfectants.

Recombinant Vectors and Virus Derivatives	Virus Family/Group	Replication-Competent	Helper Virus	Disinfectant Activity Level to be Used
Adenoviruses	*Adenoviridae*	yes		*limited spectrum of virucidal activity*
		no	Adenoviruses	*limited spectrum of virucidal activity*
Herpes viruses	*Herpesviridae*	yes		*active against enveloped viruses*
Measles virus	*Paramyxoviridae*	yes		*active against enveloped viruses*
AAV vectors (all AAV types)	*Parvoviridae*	no	Baculoviruses	*virucidal activity PLUS*
			Adenoviruses	*virucidal activity PLUS*
			Herpes viruses	*virucidal activity PLUS*
			Helper virus-free system for the production of AAV vectors	*virucidal activity PLUS*
Vaccinia virus	*Poxviridae*	yes		*active against enveloped viruses*
Lentiviral vectors (HIV)	*Retroviridae*	yes		*active against enveloped viruses*
		no		*active against enveloped viruses*
MuLV vectors and other retroviral vectors	*Retroviridae*	yes		*active against enveloped viruses*
		no		*active against enveloped viruses*
VSV	*Rhabdoviridae*	yes		*active against enveloped viruses*

AAV, adeno-associated virus; MuLV, murine leukaemia virus; VSV, vesicular stomatitis virus.

## Data Availability

Not applicable.

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
