# Peer review of "Suitable Disinfectants with Proven Efficacy for Genetically Modified Viruses and Viral Vectors"

_viruses, 2023, doi:10.3390/v15112179_

Round 1
Reviewer 1 Report (New Reviewer)
Comments and Suggestions for Authors
The authors present a review of the European standards for applications and recommendations for targets for GMO's a growing concern within the broader communities. It is important to be able to demonstrate safety standards and efficacy of disinfectants for two reasons, first, protect public health, second have a demonstrable standard on which research and health organizations can rely on. Overall, the article is well written. The background is sufficient and explains state-of-the-art standards. The rationale provided for the current testing surrogates (vaccinia, VSV, etc.) is also clear. What's not clear is why GMO's would be considered as needing a higher level of scrutiny, which seems to be a main message of the manuscript. If HIV can be disinfected, why wouldn't a gene therapy lentivirus also be susceptible? The envelopes are lipids, which would be susceptible to the same detergents. Perhaps a few more sentences in the introduction regarding public perception and a growing trend of skepticism towards research institutions regarding biosafety in the wake of COVID-19, would add impact.
Comments on the Quality of English LanguageThe manuscript is well-written, but could use another round of proof-reading for grammar accuracy and word usage.
Author Response
Thank you, we have added the following sentence “Therefore a lentiviral vector in gene therapy should be as susceptible to disinfectants as HIV” on page 9. Additionally, a sentence regarding public perception and growing trend of skepticism towards research institutions regarding biosafety was added into the introduction section.
Reviewer 2 Report (New Reviewer)
Comments and Suggestions for Authors
This is article is a review of disinfectants suitable for an array of pathogens ranging from enveloped viruses to high resistant non-enveloped viruses and genetically modified viruses. The article also delves into the difficulties of the validation testing algorithms for disinfectants being more suitable for industry, hospital, or day-to-day consumer life rather than for planned use of the disinfectants in a laboratory setting. Finally, the paper discusses differences between European standards and standards used in Germany and a proposed new category of disinfectants titled "virucidal activity PLUS" for highly resistant viruses.
Overall, I think this a solid review, especially from section 3 to the end where the data and arguments are presented in a logical manner. The introduction and section 2 could benefit from additional editing and reorganization basically using the conclusion section as the framework.
1) Lines 65 - 77 of the introduction are a mix of numbered and dashed bullets with an interstitial sentence that left this reviewer somewhat confused.
2) While interesting detailed information about some of the uses of GMOs, I recommend that the information from lines 82 - 105 be shortened. That much detail of uses of GMOs distracts from the points of the review.
3) Line 106 - can either delete "The used" or move "used" to after "viruses".
4) Line 163-4 - I applaud the use of "mutatis mutandis".
5) Lines 168-170 - The authors use parvovirus as an example of a virus requiring the new proposed category "virucidal activity PLUS", but the language in lines 171-6 and in Table 4 only mention parvoviruses for this category. Do the authors expect viruses from other families to be included in this category? If so, an additional line of narrative to reinforce that point would be helpful to the reader.
6) Lines 225-30 - what data suggests the more extensive testing required in Germany compared to the rest of Europe is needed? Would be interesting and valuable to know what gaps in the current regulations are being mitigated by extensive additional testing.
7) Section 6 is outstanding.
8) Line 376 - add "are" between "there" and "also".
Author Response
1) Lines 65 - 77 of the introduction are a mix of numbered and dashed bullets with an interstitial sentence that left this reviewer somewhat confused.
ANSWER: Thank you, the mix of numbered and dash bullets have been converted into a continuous text.
2) While interesting detailed information about some of the uses of GMOs, I recommend that the information from lines 82 - 105 be shortened. That much detail of uses of GMOs distracts from the points of the review.
ANSWER: Thank you, the passage has been shortened to “Other gene therapy treatments have been designed to treat haemophilia A, haemophilia B, or patients with inherited rare diseases [8,9].”
3) Line 106 - can either delete "The used" or move "used" to after "viruses".
ANSWER: Thank you, it has been changed accordingly, “The used” has been delated.
4) Line 163-4 - I applaud the use of "mutatis mutandis".
ANSWER: Thank you.
5) Lines 168-170 - The authors use parvovirus as an example of a virus requiring the new proposed category "virucidal activity PLUS", but the language in lines 171-6 and in Table 4 only mention parvoviruses for this category. Do the authors expect viruses from other families to be included in this category? If so, an additional line of narrative to reinforce that point would be helpful to the reader.
ANSWER: Thank you, in line 168 parvovirus - the test virus - has been changed to parvoviridae to clarify that the family of parvoviruses with the activity level "virucidal activity PLUS" is meant. The authors do not currently expect viruses from other families to be included in this category.
6) Lines 225-30 - what data suggests the more extensive testing required in Germany compared to the rest of Europe is needed? Would be interesting and valuable to know what gaps in the current regulations are being mitigated by extensive additional testing.
ANSWER: The use of AAV in gene therapy has already been implemented in the German requirements for disinfectant testing through the use of the resistant test organism Murine parvovirus. This test virus and the virus claim are also a wish of the European pharmaceutical industry and should therefore also be adopted in the European disinfectant testing. This would mean that the stricter requirements already implemented in Germany would also apply in other European countries.
7) Section 6 is outstanding.
ANSWER: Thank you.
8) Line 376 - add "are" between "there" and "also".
ANSWER: Thank you, “are” has been added.
This manuscript is a resubmission of an earlier submission. The following is a list of the peer review reports and author responses from that submission.
Round 1
Reviewer 1 Report
Comments and Suggestions for Authors
This paper attempts to provide an overview of the classification of disinfectants for viruses, comparing EU and German standards and introduces the need for a newly proposed category – virucidal activity PLUS.
Some specific points:
Line 98: the comma after ‘for’ should be removed.
Lines 139-141: Please changed the word ‘virion’ to ‘virus’ when referring to the forms of VACV.
Paragraph beginning Line 124: This is very poorly written, lacks clarity in structure and is somewhat repetitious.
Line 166/7: Do you have a reference for this?
Line 215: ‘from’ not ‘of’
Beginning of section 4 repeats what has already been said.
Tables could be better presented, e.g. ‘Activity level’ is in the legend, it doesn’t need to be in the table, which would make the basis of declaration more clearer as headings/categories.
Line 249: should be ‘poliovirus’, also in Table 3.
Line 253: “probably responsible” – it is unclear to me how this is relevant to this article, especially when phrased like speculation.
Line 279: should be ‘routine’ not ‘routinely’.
Line 298: should be ‘intrinsically’.
Line 331: explaining this regulation and just what products are affected a bit better might be of worth.
Line 333: Sentence beginning “with”- this sentence is poorly written and unclear. Please modify.
Line 337: ‘for centuries’ – sounds like exaggeration.
There is random capitalisation of words in some headings.
Comments on the Quality of English LanguageOverall, the paper is quite poorly constructed, and I recommend having it looked at by an English editing service. There is significant repetition of ideas/points, redundant sentences, and I found the flow and argument presentation disjointed.
Reviewer 2 Report
Comments and Suggestions for Authors
The review article "Suitable disinfectants with proven efficacy for genetically modified viruses and viral vectors” by Eggers et al. is well written and quite clear to follow. It provides a quite vast description of the criteria of evaluation of carrier test for virucidal activity for the different structural requirements of viruses. The manuscript misses to add a proper explanation of the differences between genetic modified viruses, oncolityc viruses, viral vectors and the virus family they belong to. It is very difficult without any detail to understand why an oncolytic herpesvirus should be different from a field herpesvirus from the point of view of its susceptibility to disinfectants. The important point in here is the envelope, therefore, a lipid bilayer. The same applies for any other modified virus, unless the authors provide examples in which the modification of the virus has led to a completely different external surface. This could well be the case of some viruses without envelope, where the modifications of the capsid proteins can eventually lead to the production of viral based vectors, a sort of viral cages, with a completely different symmetry, therefore with different gaps generated between capsomers that may increment or reduce the possibility of disinfectant to act. But this needs to be described somewhere in the review otherwise the principal point, also stated into the title, is completely untreated into the review. If literature data are not enough to reach conclusion similar to the hypothesis I forewarded, then the review need to change name and be completely reformulated from its foundation. A suggestion would be to plan a review treating the interesting subject of the so called “carrier test” and analyse the different methods available and in particular to focus on the environmental assets of the different condition we can find these viruses in real life. This would add a lot of information for the reader.
Comments on the Quality of English Language
The English language is good
Round 2
Reviewer 1 Report
Comments and Suggestions for Authors
The changes made have improved the quality of the paper.
Comments on the Quality of English LanguageThe paper has improved but the english quality still needs improvement. You state it has been checked by a professional language editor - I would be asking for my money back. At the very least I recommend re-reading the paper again very carefully to correct a few grammar and punctuation errors that still remain. Also, I feel having two consecutive sets of dot points in the introduction is not ideal.
Author Response
Meanwhile, the document has undergone English language editing by MDPI.

Reviewer 2 Report
Comments and Suggestions for Authors
I have no further suggestions compared to the ones already provided in the first review
Comments on the Quality of English LanguageThe english language is fine.
Author Response
Thank you so much!